# Impacts of sediment derived from erosion of partially-constructed road on aquatic organisms in a tropical river: The Río San Juan, Nicaragua and Costa Rica

Blanca Ríos Touma [1,2], G. Mathias Kondolf [3,4]*, Scott Walls[3]

1 Center for Environmental Design Research, University of California, Berkeley, California, United States of America, 2 Facultad de Ingenierías y Ciencias Aplicadas, FICA- Grupo de Investigación en Biodiversidad, Medio Ambiente y Salud -BIOMAS-, Universidad de Las Américas, Quito, Ecuador, 3 Department of Landscape Architecture and Environmental Planning, University of California, Berkeley, California, United States of America, 4 Collegium de Lyon—Institut des Etudes Avancées de l'Université de Lyon, Lyon, France

* kondolf@berkeley.edu

**Data Availability Statement:** All relevant data are within the manuscript and its Supporting Information files.

## Abstract

Throughout the humid tropics, increased land disturbance and concomitant road construction increases erosion and sediment delivery to rivers. Building road networks in developing countries is commonly a priority for international development funding based on anticipated socio-economic benefits. Yet the resulting erosion from roads, which recent studies have shown result in at least ten-fold increases in erosion rates, is not fully accounted for. While effects of road-derived sediment on aquatic ecosystems have been documented in temperate climates, little has been published on the effects of road-induced sediment on aquatic ecosystems in developing countries of the tropics. We studied periphyton biomass and macroinvertebrate communities on the deltas of Río San Juan tributaries, comparing north-bank tributaries draining undisturbed rain forest with south-bank tributaries receiving runoff from a partially-built road experiencing rapid erosion. Periphyton biomass, richness and abundance of macroinvertebrates overall, and richness and abundance of Ephemeroptera, Plecoptera and Trichoptera were higher on the north-bank tributary deltas than the south-bank tributary deltas. These findings were consistent with prior studies in temperate climates showing detrimental effects of road-derived fine sediment on aquatic organisms. A Non-Metric Multidimensional Scaling (NMDS) analysis showed the impacted community on the south-bank deltas was influenced by poorly-sorted substrate with greater proportions of fine sediment and higher water temperatures.

## Introduction

When released in rivers and streams, human-caused sediment can induce changes to the physical habitat and aquatic biota downstream of the sediment source [1–3]. Habitat modifications include changes from larger, more stable substrates, to smaller, unstable substrates. The

**Funding:** Field work for data collection by BRT, GMK, and SPW was supported by the Embassy of Nicaragua in the Hague, Netherlands. Manuscript preparation for BRT, GMK, and SPW was supported by the Beatrix Farrand Fund of the Department of Landscape Architecture and Environmental Planning of the University of California Berkeley, and manuscript preparation for GMK was also supported by the Collegium de Lyon - Institut des Etudes Avancées de l'Université de Lyon, the EURIAS Fellowship Programme and the European Commission (Marie-Sklodowska-Curie Actions - COFUND Programme - FP7) (no grant numbers). The funders had no role in study design, data collection and analysis, decision to publish, or preparation of the manuscript.

**Competing interests:** The authors have declared that no competing interests exist.

resulting increase in suspended sediment concentrations and turbidity can impair respiration in fish and invertebrates. Increased sedimentation can also affect primary producers at the base of the food chain by a) reducing light penetration with a resulting reduction in primary productivity [4]; b) reducing the organic content of periphyton cells [5,6]; c) abrading and damaging macrophytes [7,8]; and d) preventing attachment to substrate and removing periphyton and aquatic macrophytes in extreme events [9].

While stream biota may be adapted to variability in flow and sediments, artificially-elevated sediment inputs can have severe effects on the benthic macroinvertebrate communities, including drift due to unstable substrate, reduction of suitable habitat for some species [10], reduction of respiration due to silt deposition on breathing structures or oxygen reduction [11], changes in food availability [5,6,12], and overall changes in the river food web [13]. Increased fine sediment loads can disproportionately affect macroinvertebrates of long-lived forms, those that feed by scraping, and those that cling on substrates, as documented by Richards et al. [10].

Studies linking increased sediment loading to aquatic ecology have been conducted primarily in temperate climates. However, throughout the humid tropics, increasing land disturbance from accelerated development (e.g., mining, agricultural expansion, timber harvest) and concomitant road construction can be expected to increase erosion rates and sediment delivery to rivers [14,15]. Building road networks in developing countries is commonly a priority for international development funding, based on anticipated socio-economic benefits, but without fully accounting for the resulting increased hillslope erosion rates and induced deforestation [16]. Studies of rapid expansion of the road network in recent decades in southeast Asia have demonstrated significant ecological fragmentation [17] and at least ten-fold increases in erosion rates [18] from roads, but there has been little published on the effects of road-induced sediment increases on aquatic ecosystems in developing countries of the tropics.

The effects of deforestation and conversion to agriculture on aquatic communities in receiving waters have been documented in the tropics [e.g., 14]. Regarding road-derived sediment impacts on aquatic ecology, we found multiple studies from temperate regions, but only one such study in a neotropical river. Fossati et al. [19] documented the effects of increased sediment loads from road construction on epibenthic gatherers (e. g. Ephemeroptera: Leptohyphidae, Coleoptera: Elmidae), swimmers (Ephemeroptera: Leptophlebiidae), and scrapers (Coleoptera: Psephenidae, Trichoptera: Hydroptilidae) of the Río Coroico in the humid Yungas Mountains of the Bolivian Andes [19]. The Coroico has naturally clear water at low flow, but road construction resulted in a 500-fold increase in suspended sediment loads downstream of the disturbance. In turn, this produced a 200-fold decrease in macroinvertebrate abundance, and a 6-fold decrease in number of taxa [19].

Since 2010, efforts to construct a road along the south bank of the Río San Juan in Costa Rica have resulted in highly-visible inputs of sediment, providing an opportunity to assess potential sediment impacts on the riverine ecosystem. The objective of this study was to assess the impact of these large sediment inputs on benthic communities through the study of periphyton biomass and macroinvertebrate assemblages to enhance the knowledge of this widespread impact.

## Study area

The Río San Juan begins at the outlet of Lake Nicaragua and flows approximately 200 kilometers (km) eastward to the Caribbean Sea (Fig 1), dropping 32.7 meters (m) from the lake to the sea. About 30 km upstream of the mouth, the river splits into two distributaries, the Lower Río San Juan, and the larger Río Colorado. The border between Nicaragua and Costa Rica lies about 5 km south of the south shore of Lake Nicaragua and 5 km south of the Río San Juan for

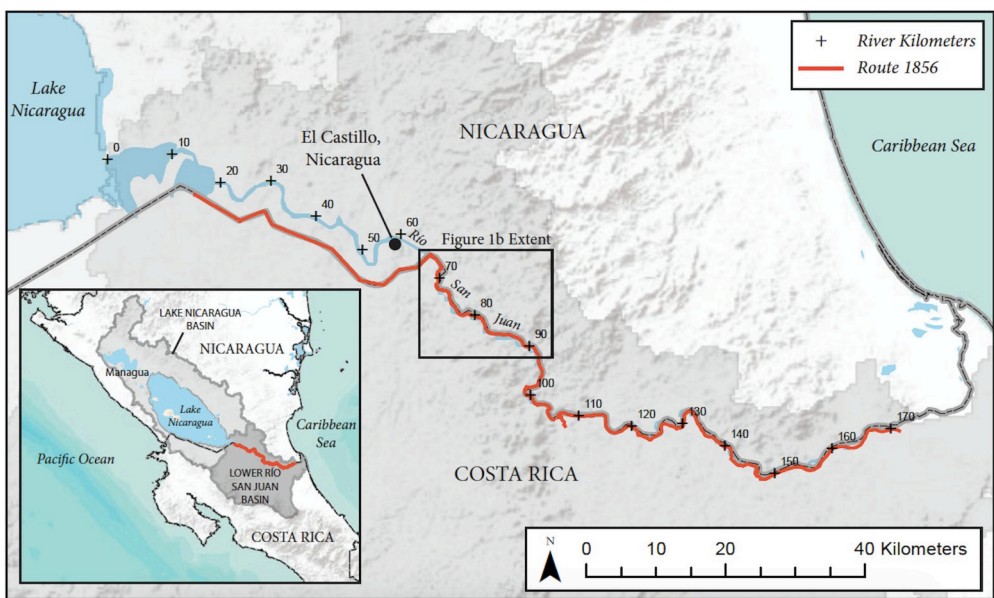

**Fig 1. Location map, Río San Juan.** Río San Juan basin in its entirety and Río San Juan from Lake Nicaragua downstream to the Caribbean Sea.

about the river's first 65 km. Then for about 135 km, the border is the south bank of the Río San Juan, until about 2 km above the mouth (from which point the exact boundary between Costa Rica and Nicaragua has been in dispute).

The lower 135 km of the river is flanked on the north bank by the Indio Maiz Biological Reserve, an area of over 4,500 km$^2$ of protected, primary tropical forest in Nicaragua. The tributary basins along the north side of this part of the river are generally small, not exceeding 418 km$^2$, and mostly drain the intact rainforest.

By contrast, the south bank of the river has been colonized by small farms, and some of the forest has been cleared for pastures. The south-bank tributaries include some much larger basins, including the Río San Carlos (2640 km$^2$) and Río Sarapiqui (2770 km$^2$), both of which experienced massive deforestation and conversion to chemical-intensive agriculture such as pineapple plantations from 1950 to 1995 [20,21]. The Río San Juan is famous for its fishery, which has included large marine fish such as tarpon (*Megalops atlanticus* Valenciennes 1847), migrating upstream through the Río San Juan into Lake Nicaragua [22].

Beginning in late 2010, the Costa Rican government began construction of a 160-km road, of which 108 km is located along the south bank of the Río San Juan, down to the bifurcation of the Río Colorado. The road was designated as the "Juan Rafael Mora Porras Route 1856" (hereafter, "Rte 1856"). Multiple contractors were hired to work on different sections of the road. At present, only discontinuous sections of the road have been completed, and large areas destabilized by bulldozers remain exposed to soil erosion and mass wasting (Fig 2).

Of the 108 km along the river's south bank disturbed for the road, 49.5 km are within 100 m of the riverbank, and 17.9 km are within 50 m. The 41.6 km of the road's intended route upstream of the confluence of Río San Carlos traverses much steeper terrain than the rest of the 108 km adjacent to the river. Of this steeper part of the route, 28.3 km (68%) are within 100 m of the riverbank, and 12.3 km (30%) are within 50 m of the river. Thus, sediment eroded from road construction has only a short distance to travel before entering the Río San Juan or its tributaries. In addition to Rte 1856 itself, multiple north-south access roads were constructed, mostly following tributary channels, which created additional disturbance and increased sediment yield.

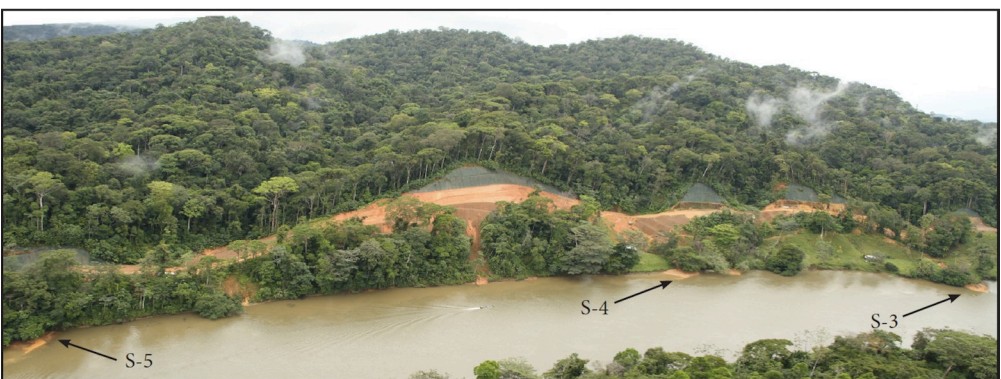

**Fig 2. Reach of Río San Juan from approximately River Km 83.3 to 84.3 (downstream of the outlet of Lake Nicaragua).** Massive cuts in steep slopes and large, eroding fill piles are visible, along with freshly deposited deltas of sediment built of road-derived sediment. The deep cut slopes and massive fill piles are subject not only to sheet erosion, but deep gully erosion and mass failures as inadequate stream culverts have blown out. Sediment is transported directly from eroding surfaces into the Río San Juan, some remaining as delta deposits. Sample sites S-3, S-4, and S-5 are identified in the photo. The boat visible in river (near center of photo) is about 7 m long. Oblique aerial view looking south by Kondolf, March 2015.

Rte 1856 was controversial, both within Costa Rica and internationally, with critical reports by the Costa Rican Federation of Architects and Engineers (CFIA) and the Costa Rican Laboratorio Nacional de Materiales y Modelos Estructurales (LANNAME) pointing to the lack of planning, poor construction practices, lack of erosion control, and excessive sediment delivery to the river and wetlands [23,24]. The road was the subject of legal cases brought before the Central American Court of Justice and the International Court of Justice, in which context the research reported here was conducted [25].

Sediments eroded from the road are carried into the Río San Juan through the over 128 mapped tributaries it crosses, through small gullies eroded on areas disturbed by road construction, or directly from disturbed slopes into the river. Most sediment enters the river and is transported downstream, but some remains visible as deposits on pre-existing natural deltas of tributaries, and sediment from some rapidly eroding sections of the road has built new deltas in locations where deltas did not previously exist (Fig 2).

## Materials and methods

### Sampling strategy

We sought to document ecological effects of sediment eroded from Rte 1856 by sampling benthic organisms, which are widely used as indicators of ecosystem health. Since they live on the benthos of the streams and rivers, their composition, richness and abundance reflect the recent history of conditions in the river, thereby providing information regarding impairment of rivers. Sampling both macroinvertebrates and periphyton is affordable and produces reliable information about water quality [26]. Macroinvertebrates are used worldwide in stream and river bio-monitoring programs [26,27]. Benthic invertebrates and algae (periphyton) are among the required indicators to establish the ecological quality according to the European Water Framework Directive [28]. Costa Rican law also requires sampling and analysis of macroinvertebrates as a basis to evaluate and classify surface water quality [29].

The protocols for sampling benthic organisms typically require collecting samples from coarse-grained substrate (gravels and cobbles) in shallow water (<0.5m deep) [30]. However, large rivers are typically too deep to meet these conditions over most of their bed, so many

macroinvertebrate sampling methods focus on sampling shallow littoral zones along the channel [31]. Most of the cross section of the Río San Juan is too deep to meet the conditions needed for shallow-water sampling, but deltas exist at the mouths of some smaller tributaries. These deltas contain gravel and cobble substrate in shallow water, and thus are suitable for colonization by macroinvertebrates and periphyton. The deltas extend from the tributary mouths, projecting into the channel from the adjacent riverbank. Thus, we sought to sample benthic communities on gravels in deltas of tributary streams, comparing conditions on deltas of streams draining undisturbed forest on the north bank (Nicaragua) with deltas affected by road-derived sediment along the south bank (Costa Rica). Deltas studied on both sides have similar ranges of drainage areas, but with the north bank including more large drainage areas and the south bank including some small drainages.

Differences in the benthic communities sampled on the two banks of the river should reflect effects of the elevated sediment loads coming from erosion of Rte 1856, as well as impacts of deforestation and use of pesticides and herbicides in the catchments draining to the deltas. We collected replicate samples in multiple deltas on both sides of the river, but we did not "pair" the samples *per se*. Rather, we collected samples from deltas spanning a range of possible drainage areas on the reach of the river most affected by the road construction.

## Site selection and characterization

As noted above, we sampled shallow-water areas on deltas, which provide habitat for the periphyton and macroinvertebrates on which water quality assessments are often based, and which also provide important habitat for juvenile fish and amphibians (although we did not sample for these organisms).

We selected 16 sites suitable for sampling of benthic indicators along the deltas of eight streams along the north bank of the Río San Juan draining undisturbed forest, and deltas of eight streams along the south bank, all of which were affected by runoff from areas disturbed in attempts to construct Rte 1856 (Fig 3). We designated north-bank sites as N-1, N-2, etc, and south-bank sites as S-1, S-2, etc. (Coordinates for each sample site are shown in Supporting Information S1 Table). Most sites were on deltas of streams with drainage areas less than 170 hectares, but sites N-1, N-3, N-8, and S-7, were on deltas of tributaries with larger drainage areas (Table 1). We conducted our sampling under the auspices of the relevant authorities, the Nicaraguan Ministry of the Environment and Natural Resources (MARENA) and the Ministry of Foreign Affairs, with staff of the agencies present during the sampling. No formal permits *per se* were required in this context.

We analyzed land cover for the catchments of each sampled tributary delta using SPOT6 satellite imagery captured February 2016. In limited areas where this satellite imagery did not cover the entire catchment, we used the most recently captured cloud-free Google Earth imagery.

We categorized land cover into five types that could be delineated using the satellite imagery. The land cover types are *Forest*, *Pasture*, *Shrub*, *Road (Rte 1856)*, and *Access Road* (roads connecting the interior to Rte 1856). We then digitized land cover polygons in ArcGIS.

We delineated catchment boundaries and drainage areas using a georeferenced 1:50,000 topographic maps from the US National Imagery and Mapping Agency (NIMA, known since 2004 as the National Geospatial-Intelligence Agency). The maps were created by the Army Corps of Engineers in 1966 from photogrammetric maps prepared by the Nicaraguan Direccíon General de Cartografía from 1960 aerial photography. We supplemented the topographic maps with the SPOT6 imagery as well as oblique aerial photography captured from helicopter to refine the delineations, particularly in smaller catchments. We then calculated the areas for each land cover type contributing to each tributary delta using ArcGIS.

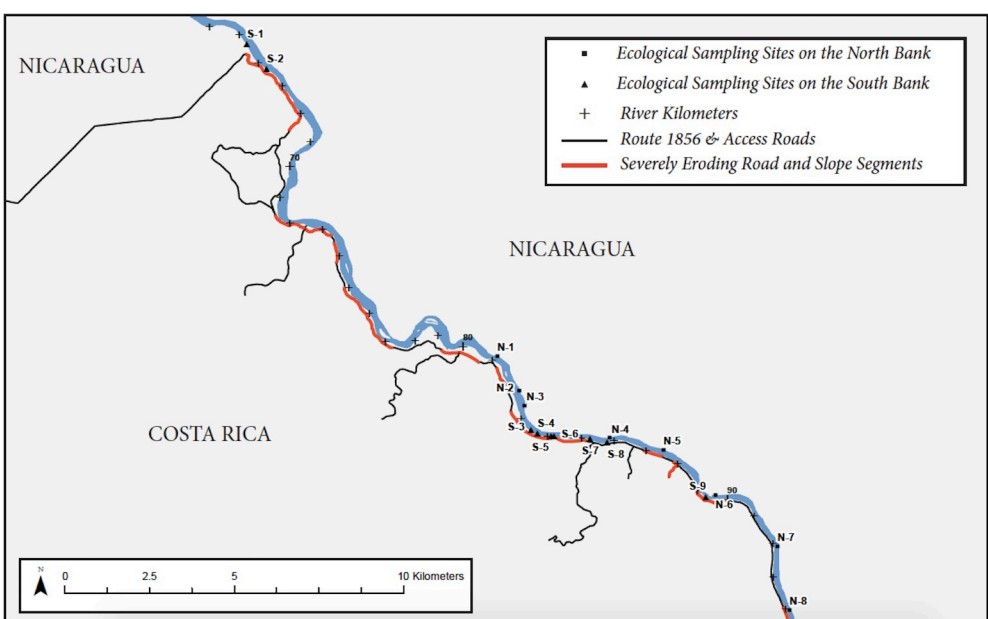

**Fig 3. Location map of sample sites.** Detail of sampled reach, showing river downstream of Lake Nicaragua outlet, areas of severe erosion due to disturbance from road construction, and locations of sample sites.

**Table 1. Land uses of sampling sites along the South and North Banks of the Río San Juan (Drainages unnamed except as indicated).**

| | Site ID | RKM[1] | Forested | Pasture | Access Road Corridor | Area Disturbed by Rte 1856 | TOTAL |
|---|---|---|---|---|---|---|---|
| | | | Sampling site watershed land use areas (hectares) | | | | |
| South Bank | S-1 | 65.3 | 58.6 | | | 3.8 | 62.4 |
| | S-2 | 66.3 | 140.9 | 1.7 | | 0.7 | 143.3 |
| | S-3 | 83.5 | 11.5 | 0.2 | | 0.6 | 12.2 |
| | S-4 | 83.7 | 5.5 | | | 1.5 | 7.0 |
| | S-5 | 84.1 | 36.0 | | | 0.8 | 36.8 |
| | S-6 | 84.2 | 25.9 | | | 0.3 | 26.2 |
| | S-7* | 85.3 | 608.5 | 2.8 | 8.4 | 0.1 | 619.7 |
| | S-8 | 85.8 | 115.1 | 10.8 | 7.0 | 0.4 | 133.1 |
| | Average South Bank | | | | | | 130.1 |
| North Bank | N-1+ | 81.1 | 6134.8 | | | | 6134.8 |
| | N-2 | 82.2 | 130.0 | | | | 130.0 |
| | N-3 | 82.6 | 1561.8 | | | | 1561.8 |
| | N-4 | 85.9 | 34.7 | | | | 34.7 |
| | N-5 | 87.4 | 165.2 | | | | 165.2 |
| | N-6 | 89.6 | 18.1 | | | | 18.1 |
| | N-7 | 92.1 | 124.6 | | | | 124.6 |
| | N-8 | 94.1 | 269.2 | | | | 269.2 |
| | Average North Bank | | | | | | 1054.8 |

[1]Kilometers measured in downstream direction along the right bank beginning at the outlet of Lake Nicaragua / Río Frio confluence.

[2]sg = $(d84/d16)^{1/2}$.

* Caño Venado.

+ Río Samoso.

We collected samples from the tributary deltas three times during spring 2014: in late March, mid-April, and early May. We sampled each delta one time during each of the three sampling campaigns. To characterize the sites, we measured temperature, pH and conductivity with field probes. We also conducted pebble counts [32] to characterize grain size of the sites.

## Benthic periphyton sampling and analysis

At each of the 16 sample sites, we sampled the periphyton biomass on similar substrate (pebbles and cobbles), according to Steinman et al. [33], scraping a fixed area (4x4 centimeters) of three different cobbles or pebbles. We then filtered the samples in a Whatman® glass microfiber circle filters, Grade GF/F (47 millimeters). The filter was stored on a glass container covered by aluminum paper and stored at 4°C when in transport (maximum 4 hours) and then stored at -20°C until the analysis in the laboratory. The analysis included the extraction in 15 milliliters (mL) of 90% buffered acetone for 24 hours in the dark, centrifugation and then measurements of chlorophyll *a* in a spectrophotometer. Living algae contain mainly undegraded chlorophyll, but with algal senescence or death, detritus degradation products also appear in the samples, mainly pheophytin [33]. Because pheophytin absorbs light in the same spectrum of chlorophyll *a*, measurements have to be corrected by acidifying the samples (with 0.1 mL of 0.1N HCL for 3 minutes), making measurements before and after the acidification.

Turbidity and colored materials can interfere with chlorophyll *a* measurement [33]. To correct the chlorophyll *a* values for the effects of turbidity and colored materials we subtracted the absorption readings at 750 nanometers (nm) from those at 664 nm. For the pheophytin correction, after acidifying the sample, we subtracted the absorption readings at 750 nm from those at 665 nm (for turbidity correction purposes).

We calculated chlorophyll *a* biomass using the formula:

$$Chlorophyll\ a\ (\mu g/cm^2) = 26.7(E_{664b} - E_{665a})\ x\ V_{ext}/area\ of\ substrate\ (cm2)\ x\ L$$

Where:

$E_{664b}$ = (Absorbance of sample at 664nm) − (Absorbance of sample at 750nm) before acidification;

$E_{665a}$ = (Absorbance of sample at 665nm) − (Absorbance of sample at 750nm) after acidification;

$V_{ext}$ = Volume of 90% acetone used in the extraction (mL), in our case 15 ml;

L = length of path light through cuvette (cm), in our case 1 cm;

26.7 = absorbance correction (derived from absorbance coefficient for chlorophyll *a* at 664nm x correction for acidification).

These analyses were performed at the laboratory of Empresa Nicaraguense de Acueductos y Alcantarillados Sanitarios (ENACAL) in Managua, following the Standard Methods 10200H (2) [30].

## Macroinvertebrate sampling and analysis

We sampled macroinvertebrates with a D-net of 500 microns mesh, following Standard Methods 10500 [30,34]. We took one sample per delta, collecting from as many shallow gravel-bedded areas as was possible during a two-minute sampling period (Fig 4). The two-minute sampling period allowed us to cover almost all delta areas. We fixed the samples in the field with 90% ethanol. We analyzed samples in the laboratory to the lowest taxonomical level possible (at least family level for insects). We calculated richness of Ephemeroptera, Plecoptera and Trichoptera (EPT), a commonly-used indicator of water quality, because these three families

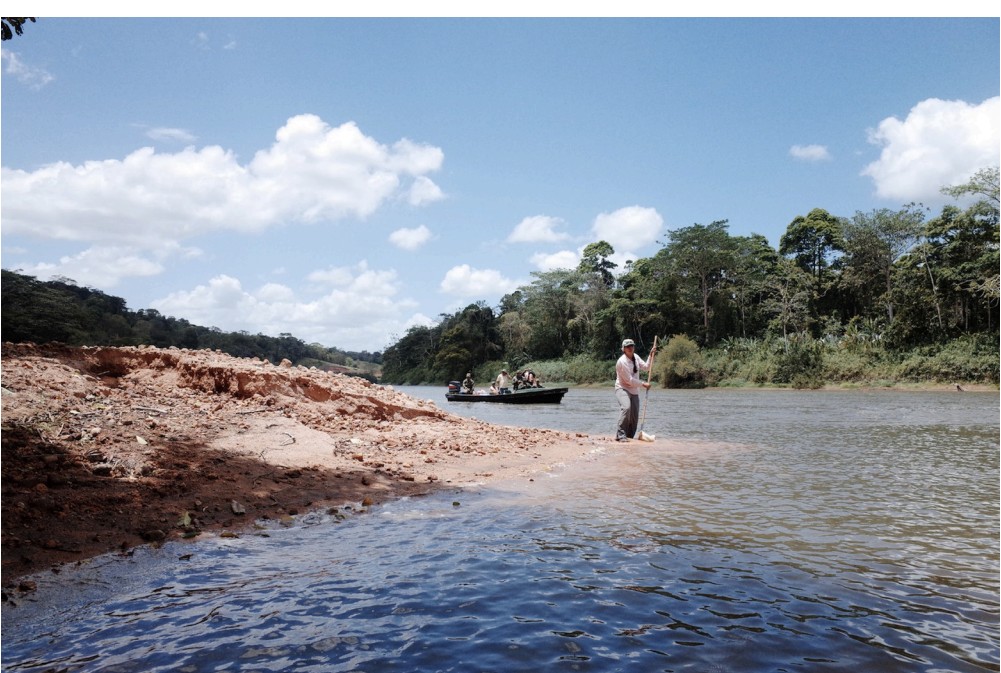

**Fig 4. Collecting benthic macroinvertebrate sample using D-net on freshly deposited delta sediments (site S-2).**
Photo looking upstream, by Walls, March 2014.

are sensitive to organic pollution [35]. The metrics to describe macroinvertebrate assemblages were richness, abundance, and EPT richness and abundance.

## Statistical analysis

To characterize substrate, we plotted cumulative size distribution curves and drew the d16, d50 and d84 values, which are the sizes at which 16, 50 and 84% of the sampled sediments are smaller, respectively. The d50 is the median size, i.e., half of the grains in the sample were larger, half smaller; it is a commonly used indicator of central tendency of the size distribution [36]. Sorting refers to the extent to which the sediments are of similar size and reflects the processes of selective transport and deposition of sediments by river flows. Sediments that have been subject to fluvial transport for a longer period tend to be better sorted than sediments recently derived from erosion of bedrock, landslides, and debris flows, which tend to have a wider range of grain sizes present. To assess how well sorted the gravels were, we calculated the geometric sorting coefficient [37,38] as $sg = (d84/d16)^{1/2}$, where the smaller the coefficient, the better sorted the sediment. To compare environmental variables between deltas draining the road and deltas of creeks draining forest we used the Median test (Chi square).

To analyze differences in periphyton biomass and macroinvertebrate metrics between deltas draining the road versus those draining forest, we used Analysis of Covariance (ANCOVA). We used our biological metrics as the response variable, riverbank as an independent variable (south-bank tributaries draining lands affected by the road construction, and north-bank tributaries draining undisturbed forest). To consider differences in tributary drainage area of each sampled delta we considered drainage area and its percentage covered by forest as covariates. To achieve the parametric assumptions for ANCOVA we transformed all variables to "logx+1". After transformation, all the variables met the assumptions. We consider a significant result with a *p-value <0.05* but we used the FDA

correction to correct all *p-values* [39] and used STATISTICA software for these analyses. We also performed a Non-Metric Multidimensional Scaling (NMDS) fitting the environmental and substrate size statistics as vectors to assess differences in composition of the macroinvertebrate community [40,41].

## Results

### Substrate and environmental variables

Temperature was significantly higher at deltas of the south bank (27.27˚C, Chi-Square = 9.0, df = 1, p = 0.0027) compared to the north bank (25.9˚C). Mean conductivity was higher at south-bank sites, but the difference was not significant (Table 2).

The substrate statistics d16 and d84 differed between deltas on the north bank and the south bank. The north bank had higher d16 values than south bank (9.8 v. 8.5) (Chi-Square = 6.35, df = 1, p = 0.0117), indicating more fine sediment present in the south-bank sites. The north bank had smaller d84 values than the south bank (30.6 vs. 34.0) (Chi-Square = 4, df = 1, p = 0.0455), which combined with the higher d16 values would suggest better sorted populations on the north-bank deltas, reflecting more fluvial sorting for these features associated with natural drainages. This is consistent with the fact that much sediment on south-bank deltas was transported only short distances from rapidly-eroding, road-construction-disturbed slopes, so we would expect less sorting in the south-bank deposits. Sorting coefficients (sg) were higher on the south bank (averaging 2.0 in contrast to 1.8 on north-bank sites), indicating slightly greater dispersion (less sorted), but the difference was not significant.

**Table 2. Substrate and environmental characteristics of sampling sites on the Río San Juan.**

| | Site ID | RKM[1] | \multicolumn Substrate and environmental characteristics of deltas sampled | | | | | | |
|---|---|---|---|---|---|---|---|---|---|
| | | | d16 (mm) | d50 (mm) | d84 (mm) | sg[2] | Temperature (˚C) | pH | Conductivity (μS/cm) |
| **South Bank** | S-1 | 65.3 | 7 | 7 | 7 | 1.0 | 26.9 | 7.8 | 185.2 |
| | S-2 | 66.3 | 7 | 11.5 | 19.5 | 1.7 | 28.0 | 7.7 | 226.7 |
| | S-3 | 83.5 | 7 | 14.9 | 42 | 2.4 | 27.4 | 6.9 | 100.3 |
| | S-4 | 83.7 | 7 | 17.3 | 48.7 | 2.6 | 27.9 | 6.8 | 133.3 |
| | S-5 | 84.1 | 8 | 13.5 | 32 | 2.0 | 27.1 | 6.8 | 51.2 |
| | S-6 | 84.2 | 8.8 | 15.4 | 38 | 2.1 | 26.3 | 6.9 | 59.4 |
| | S-7* | 85.3 | 8.8 | 17.3 | 48.7 | 2.4 | 27.1 | 7.3 | 76.0 |
| | S-8 | 85.8 | 14 | 21.6 | 36 | 1.6 | 26.8 | 7.3 | 126.5 |
| | **Average South Bank** | | *8.5* | *14.8* | *34.0* | *2.0* | *27.2* | *7.2* | *119.8* |
| **North Bank** | N-1+ | 81.1 | 10.8 | 17.5 | 31 | 1.7 | 26.3 | 7.1 | 74.7 |
| | N-2 | 82.2 | 11 | 25.5 | 44.5 | 2.0 | 26.2 | 7.2 | 103.2 |
| | N-3 | 82.6 | 9.6 | 16.5 | 31 | 1.8 | 27.1 | 7.1 | 103.7 |
| | N-4 | 85.9 | 9.6 | 13.8 | 21.7 | 1.5 | 26.0 | 7.3 | 94.0 |
| | N-5 | 87.4 | 7 | 12 | 26.5 | 1.9 | 25.7 | 7.4 | 65.5 |
| | N-6 | 89.6 | 7 | 10.2 | 24 | 1.9 | 25.4 | 7.6 | 56.0 |
| | N-7 | 92.1 | 14.1 | 26.5 | 40 | 1.7 | 25.2 | 7.8 | 91.3 |
| | N-8 | 94.1 | 9.6 | 14.5 | 26 | 1.6 | 25.3 | 7.9 | 74.2 |
| | **Average North Bank** | | *9.8* | *17.1* | *30.6* | *1.8* | *25.9* | *7.4* | *82.8* |

[1]Kilometers measured in downstream direction along the right bank beginning at the outlet of Lake Nicaragua / Río Frio confluence.

[2]sg = (d84/d16)½.

* Caño Venado.

+ Río Samoso.

## Periphyton

The three sampling events at 16 sites yielded a total of 143 samples. We had to eliminate six samples due to excess of turbidity (750 nm readings higher than 664 and 665 nm readings), all from south-bank deltas. We eliminated two samples from south-bank deltas and one from a north-bank delta for pheophytin measures exceeding the chlorophyll *a* measurements, meaning that the periphyton was not alive in those samples at the moment of collection. After this first round of elimination, we had 63 samples from deltas along the south bank and 73 from deltas along the north bank. Average chlorophyll *a* was nearly two times higher on the north-bank than south-bank sites (Fig 5 and Table 3), differences that were shown to be highly significant in ANCOVA tests; watershed area and percentage of forest cover did not explain the differences observed in periphyton biomass.

## Macroinvertebrates

We found 54 groups of macroinvertebrates in the tributary deltas of Río San Juan. Macroinvertebrate richness (Fig 6 and Table 4) was significantly higher in the north-bank deltas than in the south-bank deltas.

Tributary drainage area and percentage of forest cover did not explain the differences observed in richness (Table 5). Abundance was also significantly higher in the north-bank deltas compared to the south-bank deltas, but also was higher for larger watersheds and higher percentage of forest cover (Fig 7 and Table 5). EPT richness and abundance were higher on north-bank deltas than south-bank, although the difference was not significant (Table 4).

The Non-Metric Multidimensional Scaling analysis (NMDS, overall stress = 0.185) showed (Fig 8) a segregation of the assemblages of macroinvertebrates of most north and south-bank deltas across the axis 2.

This axis had negative relationship (Supporting Information S2 Table) with d16, d50 and pH, but positive relations with water temperature, and d84, thereby indicating that macroinvertebrate composition on south bank deltas was influenced by smaller d16, d50, larger d84, and higher water temperature. The only exception was site S-5 (south bank) that clustered with points from the north bank (Fig **8**). On the other hand, the macroinvertebrate assemblages on in north-bank deltas were positively associated with larger d16, d50, lower temperatures and better-sorted sediments (lower sg coefficient).

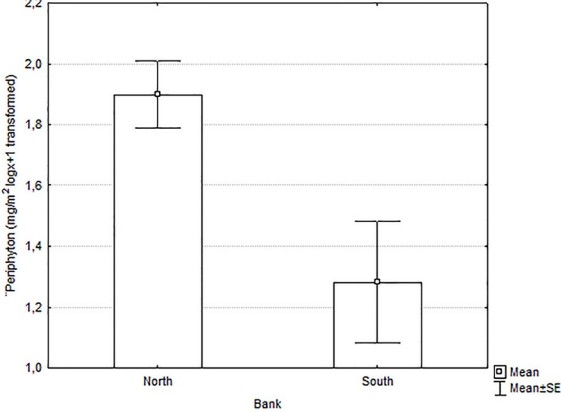

**Fig 5. Periphyton biomass (chlorophyll *a*) on benthic substrate (cobbles and pebbles) in deltas along the south versus north banks of the Río San Juan, March-May 2014.** Values are logx+1 transformed.

**Table 3. Mean, Minimum and Maximum Chlorophyll *a* Values Sampled from South (S) and North (N)-bank Sites.**

| Site | Mean (mg/cm$^2$) | Minimum (mg/cm$^2$) | Maximum (mg/cm$^2$) |
|---|---|---|---|
| S-1 | 1.75 | 0.10 | 3.40 |
| S-2 | 2.29 | 0.10 | 5.41 |
| S-3 | 1.81 | 0.20 | 5.11 |
| S-4 | 3.18 | 0.10 | 5.51 |
| S-5 | 8.92 | 4.61 | 20.73 |
| S-6 | 0.77 | 0.10 | 2.00 |
| S-7 | 1.72 | 0.20 | 4.61 |
| S-8 | 4.68 | 0.20 | 12.62 |
| N-1 | 5.02 | 3.20 | 9.51 |
| N-2 | 7.62 | 0.40 | 18.32 |
| N-3 | 3.98 | 0.20 | 18.82 |
| N-4 | 3.14 | 0.50 | 10.21 |
| N-5 | 6.01 | 0.80 | 16.92 |
| N-6 | 8.17 | 2.80 | 14.12 |
| N-7 | 6.95 | 0.40 | 17.32 |
| N-8 | 6.59 | 0.70 | 20.03 |

## Discussion

### Periphyton and macroinvertebrates, south versus north bank

Both periphyton biomass and macroinvertebrate richness and abundance were higher on north-bank tributary deltas. Moreover, the macroinvertebrate communities included more sensitive taxa on north-bank deltas than south-bank deltas. All these measures strongly indicate that conditions were more favorable to aquatic organisms along the north-bank deltas than along the south-bank deltas.

Periphyton biomass was higher in north bank deltas, independent of tributary drainage area and land cover (% of forest cover in the watershed, Tables 3 and 5). The only samples that had to be eliminated for the analysis due to excessive turbidity were from south bank, indicative of sediment impacts, consistent with results of prior studies in aquatic photosynthetic organisms exposed to abnormally high sediment loads from human activities in the catchment [7,8] or in-channel works [9].

We found significantly lower richness and abundance of macroinvertebrates in deltas draining the road (south bank) compared to deltas draining intact forest (north bank). Only abundance showed a positive relationship with tributary drainage area and percentage of forest cover (covariates in the ANCOVA analysis), meaning that abundance was also related to these variables, but the location was significant as well (south versus north bank). Diversity was significantly lower on south-bank deltas, and some taxa found on the north-bank deltas were not found on the south-bank deltas.

At least 16 EPT genera of macroinvertebrates occur in this region (Supporting Information S3 Table). As these taxa are sensitive to suspended sediment increase and fine sediment deposition (nine of them highly sensitive), they are often considered as indicators of good water quality, and sensitive to environmental changes [35,42] such as fine sediment deposition [43]. These taxa occurred with higher abundance on north-bank deltas than on south-bank deltas, although the differences were not significant.

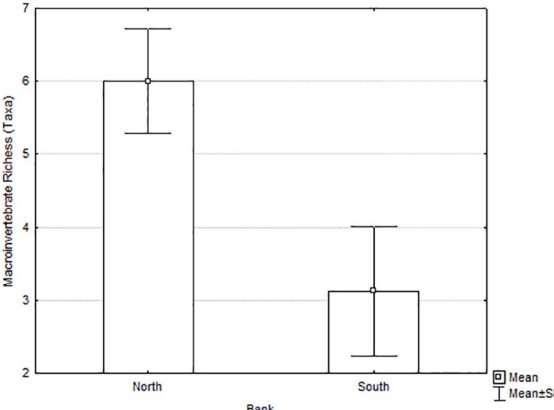

**Fig 6. Macroinvertebrate richness of south-bank versus north-bank tributary deltas of the Río San Juan, March-May 2014.**

## Distinguishing effects of road construction from other influences

The influence of substrate size on the composition of the macroinvertebrate assemblages in the NMDS analysis suggest that habitat availability and habitat quality for macroinvertebrates was largely responsible for differences in the assemblages between deltas affected by road-derived sediment versus deltas draining undisturbed forest (Fig 8). Thus, large sediment inputs are clearly implicated, but there are other factors that might negatively affect conditions on deltas draining the south bank of the river. Tributary deltas will be influenced by runoff from the entire tributary catchment. Some deltas had very small drainage areas, because they were built by sediments delivered by large gullies eroding areas disturbed for road construction.

**Table 4. Macroinvertebrate Metrics Sampled from South (S) and North (N)-bank Tributary Delta Sites on the Río San Juan.**

| Site | Richness (Av.) | S.E. Richness | Richness (min-max) | Abundance (Av.) | S.E. Abundance | Abundance (min-max) | EPT Richness (Av.) | S.E. EPT Richness | EPT Richness (min-max) | EPT Abundance (Av.) | S.E. EPT Abundance | EPT Abundance (min-max) |
|---|---|---|---|---|---|---|---|---|---|---|---|---|
| S-1 | 1.7 | 0.3 | 1–2 | 2.7 | 0.7 | 2–4 | 0 | 0 | 0 | 0 | 0 | 0 |
| S-2 | 2.3 | 0.3 | 2–3 | 5.7 | 3.2 | 2–12 | 0.3 | 0.3 | 0–1 | 0.7 | 0.7 | 0–2 |
| S-3 | 3.0 | 0.6 | 2–4 | 6.0 | 3.5 | 2–13 | 0.7 | 0.7 | 0–2 | 1.0 | 1.0 | 0–3 |
| S-4 | 5.3 | 2.3 | 3–10 | 15.3 | 10.9 | 3–37 | 1.7 | 0.9 | 0–3 | 2.7 | 1.8 | 0–6 |
| S-5 | 8.3 | 1.8 | 5–11 | 32.7 | 4.3 | 27–41 | 1.7 | 0.7 | 1–3 | 4.3 | 2.3 | 2–9 |
| S-6 | 2.0 | 0.0 | 2–2 | 5.0 | 0.6 | 4–6 | 0 | 0 | 0 | 0 | 0 | 0 |
| S-7 | 1.0 | 0.0 | 1–1 | 3.5 | 2.5 | 1–6 | 0 | 0 | 0 | 0 | 0 | 0 |
| S-8 | 1.3 | 0.3 | 1–2 | 3.0 | 1.0 | 1–4 | 0 | 0 | 0 | 0 | 0 | 0 |
| N-1 | 7.3 | 2.6 | 3–12 | 68.0 | 40.4 | 11–146 | 0.7 | 0.3 | 0–1 | 1.3 | 0.7 | 0–2 |
| N-2 | 9.0 | 3.6 | 4–16 | 24.3 | 5.8 | 15–35 | 3.3 | 1.9 | 1–7 | 9.7 | 6.7 | 2–23 |
| N-3 | 3.0 | 0.0 | 3–3 | 9.5 | 2.5 | 7–12 | 0 | 0 | 0 | 0 | 0 | 0 |
| N-4 | 6.0 | 0.6 | 5–7 | 99.3 | 52.0 | 10–190 | 1.3 | 0.9 | 0–3 | 1.7 | 0.9 | 0–3 |
| N-5 | 5.3 | 1.2 | 3–7 | 78.3 | 51.5 | 20–181 | 0.3 | 0.3 | 0–1 | 0.7 | 0.7 | 0–2 |
| N-6 | 8.0 | 1.7 | 5–11 | 30.3 | 12.5 | 15–55 | 2.3 | 0.7 | 1–3 | 8.7 | 3.9 | 1–14 |
| N-7 | 5.3 | 1.2 | 3–7 | 16.7 | 6.2 | 5–26 | 0.3 | 0.3 | 0–1 | 0.7 | 0.7 | 0–2 |
| N-8 | 4.0 | 0.6 | 3–5 | 6.7 | 1.2 | 5–9 | 0.7 | 0.7 | 0–2 | 0.7 | 0.7 | 0–2 |

Av. = Average; S.E. = Standard Error of the Mean; min-max = minimum and maximum values found.

**Table 5. ANCOVA P-values for Biological Metrics with Watershed Area, % of Catchment Forested and Bank (North and South Banks of Río San Juan, Nicaragua).**

| Factors/ Biological Metrics | Richness | Abundance | EPT Richness | EPT Abundance | Periphyton |
|---|---|---|---|---|---|
| % of catchment forested | 0,1934 | **0,0393**[1] | 0,2572 | 0,2058 | 0,6448 |
| Bank | **0,0065**[2] | **0,0009**[3] | 0,1076 | 0,0934 | 0,1399 |
| Watershed size (tributary drainage area) | 0,1573 | **0,0148**[4] | 0,2958 | 0,4500 | 0,7633 |
| Bank | **0,0491**[5] | **0,0303**[6] | 0,4824 | 0,4340 | **0,0409**[7] |

1. Significant in ANCOVA: % of forest cover had an r = 0.5 showing more abundance with higher forest cover.

2. Post-Hoc LSD test showed significant higher Richness in the North Bank compared to the South Bank in ANCOVA model.

3. Post-Hoc LSD test showed significant higher abundance in the North Bank compared to the South Bank in ANCOVA model.

4. Significant in ANCOVA: watershed size had an r = 0.7 showing more abundance at bigger watersheds.

5. Post-Hoc LSD test showed significant higher Richness in the North Bank compared to the South Bank in ANCOVA model.

6. Post-Hoc LSD test showed significant higher abundance in the North Bank compared to the South Bank in ANCOVA model.

7. Post-Hoc LSD test showed that periphyton biomass was significantly higher in the north bank independent of watershed size in ANCOVA model.

The resulting deltas are influenced primarily by the road because road runoff dominates the tributary input. However, tributary deltas with large drainage areas are exposed to more upstream influences, including increased sediment loading and contamination from agricultural chemicals. It is well established in the literature that conversion of intact rainforest to agriculture and pasture typically increases erosion rates by 10 to 100 times, but even higher rates are observed from poorly-maintained roads [44,45]. Thus, some of the impacts observed in south-bank tributaries could be attributable to land-use conversion to agriculture, but Rte 1856 and its access roads are likely more influential based on mapped land uses in the catchments (Table 1). The higher water temperatures measured in south-bank deltas are most likely attributable to effects of solar heating on deforested lands in the catchments to the south, contrasted to the forested areas on the north bank.

Some evidence with which to distinguish road effects from effects of upstream land use was provided the Centro Científico Tropical [46], which conducted a benthic macroinvertebrate sampling study in August-October 2014 in 10 south-bank tributaries to the Río San Juan crossed by Rte 1856 (Fig 9). Following protocols of MINAE [29] to assess water quality using bio-indicators, CCT collected samples upstream and downstream of the road crossing on each tributary, such that the upstream samples functioned as reference sites and the downstream

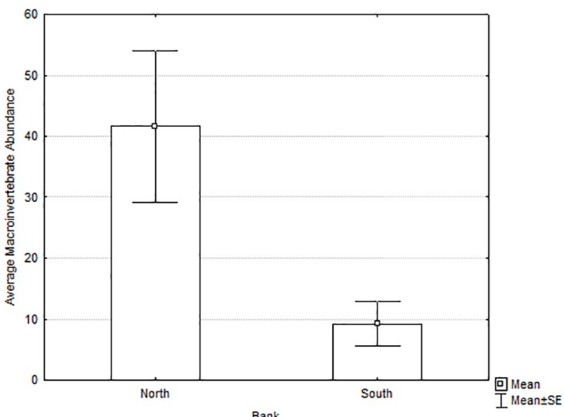

**Fig 7. Macroinvertebrate abundance of south-bank versus north-bank tributary deltas of the Río San Juan, March-May 2014.**

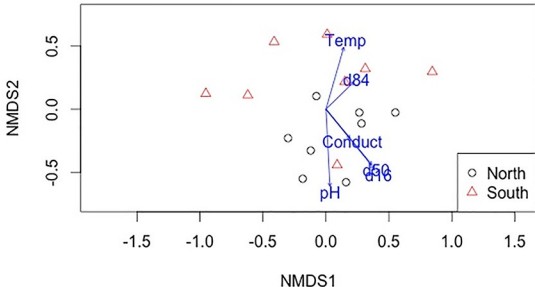

**Fig 8. Non-Metric Multidimensional Scaling (NMDS) of macroinvertebrate assemblages sampled at north-bank (circles) and south-bank (triangles) tributary deltas of the Río San Juan.** Vectors represent the substrate and environmental variables measured, fitted in the space of variation of macroinvertebrate composition. NMDS stress is 0.185.

samples as impacted sites. CCT [46] found the abundance of macroinvertebrates was lower in the sites downstream of Rte 1856 in seven of the ten streams studied, macroinvertebrate richness was lower downstream in eight of the ten streams, and the water quality (based on the Costa Rican BMWP index) was worse downstream in nine of the ten streams studied [46].

The CCT study had some methodological weaknesses, but its results can help to distinguish the effects of Rte 1856 from other factors affecting aquatic ecology of the streams draining the south side of the river. If elevated sediment loads from deforested catchments and agricultural chemicals applied to cleared lands were the principal factors affecting the macroinvertebrate communities, they should be affecting the sites upstream and downstream of the road equally. The fact that most CCT sites showed more degraded conditions below the Rte 1856 crossing provides evidence for the influence of the road, and the principal component of that runoff relevant to aquatic life is likely its high sediment load.

The negative effect of sediments eroded from the road on the aquatic communities along the south bank of the Río San Juan is consistent with patterns documented in the scientific literature from studies in rivers elsewhere [10,19,47]. Moreover, macroinvertebrate communities may be affected by the lower periphyton biomass on sediment-impacted deltas, which results in reduced food availability for macroinvertebrates, as found in previous research [5,6,12].

### Ecosystem implications

Reduced macroinvertebrate abundance and richness can have significant effects on the ecosystem, because of the importance of macroinvertebrates in the aquatic and riparian ecosystem. For example, the larval stages of aquatic insects are critically important prey for many fish, while the adult stages are important prey for birds. The effects documented here on the benthic primary producers (periphyton) could be extended up the food chain [2].

In large rivers such as the Río San Juan, tributary deltas can provide shallow, cobble-gravel habitats, along with complex features such as alcoves and cover elements such as large wood, which collectively provide important habitat diversity. Along the Río San Juan, we can expect that these deltas provide habitats for juvenile fish, as well as the periphyton and macroinvertebrates they depend upon as a food source. Comparing our results to those of the only other study of impacts of road-derived sediment on aquatic ecology in tropical Latin America [19], our results were consistent in showing negative effects of road-derived sediment, but the differences in our study were less extreme. For the Río San Juan tributaries, we have no data comparable to the 500-fold increases in suspended sediment concentrations downstream of road construction documented by Fossati et al. [19] and the resulting 200-fold decrease in

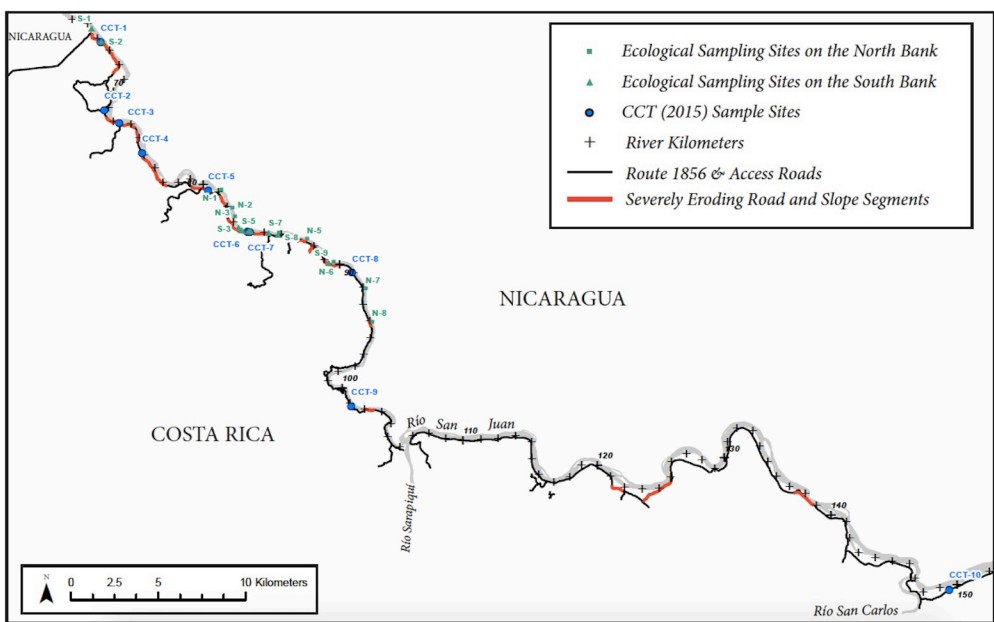

**Fig 9. Location of CCT's paired sampling sites on south-bank tributaries, upstream and downstream of the Rte 1856 crossing of the Río San Juan, showing also our sample sites for reference [46].**

macroinvertebrate abundance. For the small Río San Juan tributaries we studied, large sediment inputs are mostly in response to rain and runoff, although some chronic sediment inputs would probably persist during baseflows, because easily erodible sediment deposits would remain in contact with the flowing water. Some of the deltas we sampled, as well as the channels above and below the road crossings sampled by CCT [46], were affected by runoff from cleared pasture and access road construction in the tributary catchments, as well as runoff from Rte 1856 itself. Thus, along the Río San Juan, the baseline condition was not the extremely clear water characteristic of the Río Coroico at baseflow.

## Conclusions

Our results demonstrate that the aquatic communities on deltas along the south bank of the Río San Juan, affected by tributaries draining slopes recently disturbed for road construction, are significantly degraded compared to those developed on the deltas of tributaries entering the north bank of the river, which are not affected by the road-derived sediment. These results are consistent with results of multiple prior studies of the effect of elevated sediment loads in temperate climate streams and one prior study in the tropics. We found that periphyton biomass, and abundance and richness of macroinvertebrates, were significantly lower in the deltas receiving sediment from road construction than in deltas draining undisturbed forest. Sensitive EPT taxa were more common on the north bank than south bank, but the difference was not statistically significant. Macroinvertebrate data collected in a study conducted parallel to ours [46] indicated that the principal factor affecting aquatic organisms in tributary streams of the south bank was sediment eroded from the road, rather than runoff from their tributary catchments, some of which were affected by deforestation for access roads, industrial agriculture, and pasture.

With rapid expansion of road networks in the tropical areas, many of which are subject to high erosion rates [16], increased sediment loads are likely to have increasingly important

impacts on the aquatic ecology of receiving waters. Moreover, road networks encourage defor-estation on a massive scale, as documented across Costa Rica in the late 20th century [15,48]. Thus, in addition to the direct runoff from roads, roads have an important indirect impact on stream ecology, making deforestation more likely in the areas opened-up by the roads. A study in the Caribbean coastal region of Costa Rica indicated that streams draining deforested areas had reduced macroinvertebrate diversity and fewer sensitive taxa than streams draining for-ested areas, effects that were partially mitigated by forest buffers [14]. Further studies on the direct impacts of roads on aquatic ecosystems in tropical regions, such as the study by Fossati et al. [19] and the study presented here, are needed to understand road impacts in these here-tofore under-studied areas, where road network expansion puts important ecosystems at risk. These studies provide evidence for the imperative to avoid such impacts wherever possible, and highlight the importance of better road design and planning.

## Supporting information

**S1 Table. Coordinates of sampled deltas along the Río San Juan, Nicaragua.** South-bank tributary deltas designated as S-1, S-2, etc, north-bank deltas as N-1, N-2, etc.
(DOCX)

**S2 Table. Relations of vectors of environmental and substrate size variables with the NMDS axis.** The Non-Metric Multidimensional Scaling analysis (NMDS, overall stress = 0.185) showed a segregation of macroinvertebrate assemblages of most sampled deltas across axis 2, showing negative relations with d16, d50 and pH, but positive relations with water tempera-ture, and d84, thereby indicating that macroinvertebrate composition on south bank deltas was influenced by smaller d16, d50, larger d84, and higher water temperature.
(DOCX)

**S3 Table. Sensitivity of macroinvertebrate taxa found in the Río San Juan to suspended sediments and deposited fine sediment, based on scientific literature.** Taxa occurring in the Río San Juan identified as sensitive by Carlise *et al.* (2007) and Zweig and Rabeni (2001). Inter-mediate (or 'medium') sensitivity taxa designated as "ms"; high sensitivity taxa identified as "hs".
(DOCX)

## Acknowledgments

Raúl Acosta, University of Barcelona, identified the macroinvertebrate samples. Norlan Mejía Martinez, of the Nicaraguan Company of Sanitary Aqueducts and Sewers (ENACAL), col-lected samples in mid-April and early May 2014, and conducted laboratory analyses of the periphyton samples. Assistance in field work was graciously provided by Mario Gutiérrez Alar-cón of National Agricultural University, staff from the Nicaraguan Ministry of the Environ-ment and Natural Resources (MARENA), and ENACAL. Evanor Martinez, Universidad de León (UNAN- León), helped with macroinvertebrate sampling and sorting. Alev Bilginsoy of the University of California Berkeley assisted with manuscript preparation and submission.

## Author Contributions

**Conceptualization:** Blanca Ríos Touma, G. Mathias Kondolf, Scott Walls.

**Data curation:** Blanca Ríos Touma.

**Formal analysis:** Blanca Ríos Touma, G. Mathias Kondolf.

**Funding acquisition:** Blanca Ríos Touma, G. Mathias Kondolf.

**Investigation:** Blanca Ríos Touma, G. Mathias Kondolf, Scott Walls.

**Methodology:** Blanca Ríos Touma, G. Mathias Kondolf, Scott Walls.

**Project administration:** Blanca Ríos Touma, G. Mathias Kondolf.

**Supervision:** Blanca Ríos Touma, G. Mathias Kondolf.

**Validation:** Blanca Ríos Touma.

**Visualization:** Blanca Ríos Touma, Scott Walls.

**Writing – original draft:** Blanca Ríos Touma, G. Mathias Kondolf, Scott Walls.

**Writing – review & editing:** Blanca Ríos Touma, G. Mathias Kondolf, Scott Walls.

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
