## [Decision Letter · Decision Letter 0]

4 Aug 2020

PONE-D-20-21172

Impacts of sediment derived from erosion of partially-constructed road on aquatic organisms in a tropical river: the Río San Juan, Nicaragua and Costa Rica

PLOS ONE

Dear Dr. Kondolf,

Thank you for submitting your manuscript to PLOS ONE. After careful consideration, we feel that it has merit but does not fully meet PLOS ONE’s publication criteria as it currently stands. Therefore, we invite you to submit a revised version of the manuscript that addresses the points raised during the review process.

Both reviewers have provided specific suggestions needed to improve this paper. I agree with all of the comments and suggestions and feel that each need to be addressed. In addition, I suggest improvements and additional information are needed to improve the utility of your NMDS analysis. Further, I would suggest careful consideration is needed with regard to whether all tables and figures are presented in the best manner.

We look forward to receiving your revised manuscript.

Kind regards,

Michael A Chadwick, PhD

Academic Editor

PLOS ONE

Journal Requirements:

1. In your Methods section, please provide additional location information of the study sites, including geographic coordinates for the data set if available.

2. In your Methods section, please provide additional information regarding the permits you obtained for the work. Please ensure you have included the full name of the authority that approved the study sites access and, if no permits were required, a brief statement explaining why.

3. We note that Figure [3] includes an image of a [patient / participant / in the study]. 

Additional Editor Comments (if provided):

This is an interesting, descriptive study which address an important environmental problem which affects many developing countries. I found the work to be well written but do agree with reviewer 2 that more clarity and information is needed in the Methods to improve the paper. I also agree that providing a well defined research question and objectives in the Introduction would help readers' overall focus. A reorganization of information in the Introduction and Study Site section would also be useful too. I would suggest that map figures could be merged as they might work better as a multi-panel figure. THe NMDS results in Table 6 are redundant with the associated figure. What would have been useful to report is the overall stress of the analysis and the amount of variation explained by each axis. Finally, some of the longer tables may work better as appendices.

Reviewers' comments:

Reviewer's Responses to Questions

**Comments to the Author**

1. Is the manuscript technically sound, and do the data support the conclusions?

Reviewer #1: Yes

Reviewer #2: No

2. Has the statistical analysis been performed appropriately and rigorously? 

Reviewer #1: Yes

Reviewer #2: No

3. Have the authors made all data underlying the findings in their manuscript fully available?

Reviewer #1: Yes

Reviewer #2: Yes

4. Is the manuscript presented in an intelligible fashion and written in standard English?

Reviewer #1: Yes

Reviewer #2: No

5. Review Comments to the Author

Reviewer #1: The manuscript by Rios Touma and colleagues provides a descriptive account on the effects of erosion caused by road-development in the tributaries of the Rio San Juan (Costa Rica, Nicaragua). The study provides an assessment of the potential effects of rapid development (i.e., road construction) in a region expected to suffer from such impacts in a disproportionate manner. Furthermore, rivers in the region have seldom been studied to assess the effects of such impacts. Thus the data provided should be considered to be valuable for the continued assessment of human induced disturbances in river ecosystems of the region.

With that said, there are minor adjustments or edits that should be considered by the authors to improve the document. For example, throughout the document there are instances when orders (Ephemeroptera, Plecoptera, Trichoptera, Coleoptera) and families of macroinvertebrate taxonomic groups were provided in italics. This seems unnecessary unless these were names for genera or species (e.g., lines 42, 86-87, and elsewhere). In addition, the authors should consistently refer to the benthic community studied as 'macroinvertebrates' throughout the document and avoid referring to 'insects', unless they only assessed insects in the study (which Table 7 suggest they include other non-insect groups in their assessment).

Other minor editorial suggestions are included as attachment.

Reviewer #2: The article examines the impacts of sediment from the erosion of a partially constructed highway on aquatic organisms in a tropical river. The manuscript presents fundamental problems that go beyond the work of a review. The objectives are not well established; there is no clear research question and the sections of the manuscript are not well organized and in some cases are not well explained either. Study design is difficult to assess as information is lacking, leaving too many questions open. For these reasons, its recommendation is difficult in its current state.

Lack of information

Much information is lacking to assess study design, making it impossible to assess the quality of the data and analyzes performed.

There is no clear research question.

Lines 95-97: The authors indicate that "The objective of this study

was to assess the impact of these large sediment inputs on macroinvertebrate communities". However, the authors evaluated the periphyton and did not incorporate it into the objective of their work.

Materials and methods

"Study objectives" and "study design"

In this section the authors included the objectives, which are not clear and should also be described at the end of the introduction.

The study design is not detailed, although it is mentioned in the subtitle. Here are some questions the reader cannot answer:

When was this study done?

At what seasons of the year did they sample?

What was the frequency of sampling?

How many replicates were taken for both macroinvertebrates and the periphyton in each sample?

Finally, the authors do not show a substantial or novel contribution to what has already been previously demonstrated by other authors.

6. PLOS authors have the option to publish the peer review history of their article (what does this mean?). If published, this will include your full peer review and any attached files.

Reviewer #1: No

Reviewer #2: No

---

## [Author Response · Author response to Decision Letter 0]

13 Sep 2020

Editor Comments: Journal Requirements

1. In your Methods section, please provide additional location information of the study sites, including geographic coordinates for the data set if available.

Coordinates for sample locations are now provided in a table in the Supporting Information, S1 Table. 

2. In your Methods section, please provide additional information regarding the permits you obtained for the work. Please ensure you have included the full name of the authority that approved the study sites access and, if no permits were required, a brief statement explaining why.

Text revised to indicate that the sampling was undertaken under the auspices of the relevant authorities, with staff from these ministries present during the sampling. No formal permits were required. 

3. We note that Figure [3] includes an image of a [patient / participant / in the study]. 

The figure in the foreground is the first author, Dr Rios Touma, collecting a macroinvertebrate sample. Dr. Rios agrees to publication of the photograph. The faces of people in the background cannot be made out so our understanding is that there would be no privacy issue with using the photograph and no need to try to contact them for permission. The photo was taken in March 2014 by co-author Walls, who agrees to use of the photo in the paper. 

If there remain issues with using the photo, we can simply delete it from the paper, but we believe it is helpful to the reader’s understanding of the study to see an example of a delta on which we sampled and to see the sampling procedure. 

Additional Editor Comments 

This is an interesting, descriptive study which address an important environmental problem which affects many developing countries. I found the work to be well written but do agree with reviewer 2 that more clarity and information is needed in the Methods to improve the paper. I also agree that providing a well defined research question and objectives in the Introduction would help readers' overall focus. 

We expanded the objectives statement at the end of the Introduction.

A reorganization of information in the Introduction and Study Site section would also be useful too. I would suggest that map figures could be merged as they might work better as a multi-panel figure. 

We reviewed the sequence in which the material was presented and concluded that it is most efficient to provide the study area description before going into detailed methods, as the methods were chosen in the context of the local situation and sampling constraints. We considered the proposal to merge the map figures into a multi-panel figure, but prefer to use the map figures as is because of the challenges of managing a large multi-panel figure. While we acknowledge that such a multi-panel map figure could be very useful, we felt it would not be needed to convey the spatial relationships important for this study. However, we are open to pursuing this concept of a multi-panel further if the editors determine that doing so would be a significant improvement in the paper. 

THe NMDS results in Table 6 are redundant with the associated figure. 

We have moved Table 6 to Supporting Information. 

What would have been useful to report is the overall stress of the analysis and the amount of variation explained by each axis. 

This was originally reported in the figure legend. However, we have also added it in the text describing these results. This analysis does not provide the variation explained by each axis. Unlike PCA, this analysis represents the ordering of the observations in few directions. Our stress value was good, meaning that NMDS provided a good fit of all our sites. 

Finally, some of the longer tables may work better as appendices.

We also shifted Table 7 to Supporting Information. 

Reviewer 1 (R1) Comments 

Note that R1 also provided comments in the pdf document itself. All of R1’s proposed revisions in the document were accepted and are incorporated in the new version of the paper. 

The manuscript by Rios Touma and colleagues provides a descriptive account on the effects of erosion caused by road-development in the tributaries of the Rio San Juan (Costa Rica, Nicaragua). The study provides an assessment of the potential effects of rapid development (i.e., road construction) in a region expected to suffer from such impacts in a disproportionate manner. Furthermore, rivers in the region have seldom been studied to assess the effects of such impacts. Thus the data provided should be considered to be valuable for the continued assessment of human induced disturbances in river ecosystems of the region.

With that said, there are minor adjustments or edits that should be considered by the authors to improve the document. For example, throughout the document there are instances when orders (Ephemeroptera, Plecoptera, Trichoptera, Coleoptera) and families of macroinvertebrate taxonomic groups were provided in italics. This seems unnecessary unless these were names for genera or species (e.g., lines 42, 86-87, and elsewhere). In addition, the authors should consistently refer to the benthic community studied as 'macroinvertebrates' throughout the document and avoid referring to 'insects', unless they only assessed insects in the study (which Table 7 suggest they include other non-insect groups in their assessment).

We revised the text to eliminate the italics as per R1’s recommendation. We revised the text to clarify that our focus was on benthic macroinvertebrates and not on all insects. 

Other minor editorial suggestions are included as attachment.

All of R1’s proposed revisions in the document were accepted and are incorporated in the new version of the paper. These revisions included a statement regarding the challenges in sampling non-wadeable stream. We added a citation to a principal reference on this topic, which explains that macroinvertebrate sampling in deep rivers is often implemented along shallow littoral zones that meet the depth requirements for sampling. 

Reviewer 2 (R2) Comments 

The article examines the impacts of sediment from the erosion of a partially constructed highway on aquatic organisms in a tropical river. The manuscript presents fundamental problems that go beyond the work of a review. The objectives are not well established; there is no clear research question and the sections of the manuscript are not well organized and in some cases are not well explained either. Study design is difficult to assess as information is lacking, leaving too many questions open. For these reasons, its recommendation is difficult in its current state.

Lack of information

Much information is lacking to assess study design, making it impossible to assess the quality of the data and analyzes performed.

As detailed below, much of the information flagged by R2 as missing was included already in the paper. Below we indicate where in the original manuscript the information appeared. Regarding the frequency of sampling, we clarified that we sampled once per delta during each of the three sampling campaigns (in March, April, and May respectively). 

There is no clear research question.

Lines 95-97: The authors indicate that "The objective of this study

was to assess the impact of these large sediment inputs on macroinvertebrate communities". However, the authors evaluated the periphyton and did not incorporate it into the objective of their work.

Good point. We revised to the text to include sampling of periphyton as an objective in and of itself. Periphyton is also important as a food source for benthic macroinvertebrates.

Materials and methods

"Study objectives" and "study design"

In this section the authors included the objectives, which are not clear and should also be described at the end of the introduction.

Text revised to state the study objectives more fully at the end of the introduction. 

The study design is not detailed, although it is mentioned in the subtitle. Here are some questions the reader cannot answer:

When was this study done?

This was stated on lines 236-239 of the submitted manuscript:

“We collected samples from the tributary deltas three times during spring 2014: in late March, mid-April, and early May. To characterize the sites, we measured temperature, pH and conductivity with field probes. We also conducted pebble counts (31) to characterize grain size of the sites”

At what seasons of the year did they sample?

As per above, the sampling was undertaken in the spring, from late March to early May, which corresponds to the transition between seasonal low and high waters.

What was the frequency of sampling?

We sampled once per delta in each of the three sampling trips. To remove any ambiguity on this point, we added a sentence near the end of Methods and Materials stating, “We sampled each delta one time during each of the three sampling campaigns.”

How many replicates were taken for both macroinvertebrates and the periphyton in each sample?

For periphyton, this was stated on lines 241-243 of the submitted manuscript:

“At each of the 16 sample sites, we sampled the periphyton biomass on similar substrate (pebbles and cobbles), according to Steinman et al. (32), scraping a fixed area (4x4 centimeters) of three different cobbles or pebbles.” 

For Macroinvertebrates this was stated on lines 273-276 of the submitted manuscript:

“We sampled macroinvertebrates with a D-net of 500 microns mesh, following Standard Methods 10500 (30, 33). We took one sample per delta, collecting from as many shallow gravel-bedded areas as was possible during a two-minute sampling period (Fig 3). The two-minute sampling period allowed us to cover almost all delta areas”

Finally, the authors do not show a substantial or novel contribution to what has already been previously demonstrated by other authors.

We respectfully disagree with the reviewer. There is currently very little literature on the effects of roads in the neotropics. These largely undocumented impacts are of high relevance in Latin America, where deforestation rates are among the highest in the world, and roads play a major role in inducing deforestation. As roads commonly follow and cross rivers, their impacts on river ecosystems are undoubtedly important, but they have not been documented. Thus, our results contribute to understanding of an important issue in a region of high ecological importance.

---

## [Editor Report · Decision Letter 1]

2 Nov 2020

Impacts of sediment derived from erosion of partially-constructed road on aquatic organisms in a tropical river: the Río San Juan, Nicaragua and Costa Rica

PONE-D-20-21172R1

Dear Dr. Kondolf,

We’re pleased to inform you that your manuscript has been judged scientifically suitable for publication and will be formally accepted for publication once it meets all outstanding technical requirements.

Kind regards,

Michael A Chadwick, PhD

Academic Editor

PLOS ONE

Additional Editor Comments (optional):

Thank you for the thoughtful and rigorous treatment of the comments by all reviewers from your 1st submission. I am satisfied that you have answered all of the outstanding issue and the work is improved.
---

## [Editor Report · Acceptance letter]

9 Nov 2020

PONE-D-20-21172R1 

Impacts of sediment derived from erosion of partially-constructed road on aquatic organisms in a tropical river: the Río San Juan, Nicaragua and Costa Rica 

Dear Dr. Kondolf:

I'm pleased to inform you that your manuscript has been deemed suitable for publication in PLOS ONE. Congratulations! Your manuscript is now with our production department. 

Kind regards, 

on behalf of

Dr. Michael A Chadwick 

Academic Editor

PLOS ONE